# The effect of bacteria on planula-larvae settlement and metamorphosis in the octocoral *Rhytisma fulvum fulvum*

Isabel Freire[1], Eldad Gutner-Hoch[2¤], Andrea Muras[1], Yehuda Benayahu[3], Ana Otero[1]*

**1** Instituto de Acuicultura and Departamento de Microbiología, Facultad de Biología, Edificio CIBUS, Universidade de Santiago de Compostela, Santiago de Compostela, SPAIN, **2** School of Zoology, George S. Wise Faculty of Life Sciences, Tel–Aviv University, Ramat-Aviv, Tel-Aviv, Israel, **3** Interuniversity Institute for Marine Sciences, Eilat, Israel

¤ Current address: The Eugene Bell Center, Marine Biological Laboratory, Woods Hole, Ma, United States of America

* anamaria.otero@usc.es

**Data Availability Statement:** All relevant data are within the manuscript and its Supporting Information files.

**Funding:** This work was supported by: EU FP7-Research Infrastructure Initiative Assemble

## Abstract

While increasing evidence supports a key role of bacteria in coral larvae settlement and development, the relative importance of environmentally-acquired versus vertically-transferred bacterial population is not clear. Here we have attempted to elucidate the role of post-brooding-acquired bacteria on the development of planula-larvae of the octocoral *Rhytisma f. fulvum*, in an *in vitro* cultivation system employing different types of filtered (FSW) and autoclaved (ASW) seawater and with the addition of native bacteria. A good development of larvae was obtained in polystyrene 6-well cell culture plates in the absence of natural reef substrata, achieving a 60–80% of larvae entering metamorphosis after 32 days, even in bacteria-free seawater, indicating that the bacteria acquired during the brooding period are sufficient to support planulae development. No significant difference in planulae attachment and development was observed when using 0.45 μm or 0.22 μm FSW, although autoclaving the 0.45 μm FSW negatively affected larval development, indicating the presence of beneficial bacteria. Autoclaving the different FSW homogenized the development of the larvae among the different treatments. The addition of bacterial strains isolated from the different FSW did not cause any significant effect on planulae development, although some specific strains of the genus *Alteromonas* seem to be beneficial for larvae development. Light was beneficial for planulae development after day 20, although no *Symbiodinium* cells could be observed, indicating either that light acts as a positive cue for larval development or the presence of beneficial phototrophic bacteria in the coral microbiome. The feasibility of obtaining advanced metamorphosed larvae in sterilized water provides an invaluable tool for studying the physiological role of the bacterial symbionts in the coral holobiont and the specificity of bacteria-coral interactions.

(Association of European marine biological laboratories); EU FP7 Project Byefouling (grant agreement no 612717); Xunta de Galicia, Consellería de Cultura, Educación e Ordenación Universitaria (grant number ED431D 2017/22). The funders had no role in study design, data collection and analysis, decision to publish, or preparation of the manuscript.

**Competing interests:** The authors have declared that no competing interests exist.

## Introduction

Coral reefs are suffering substantial degradation at a global scale [1]. It is estimated that ~60% of the reefs are already daged either directly as the result of human activities or indirectly by factors derived from global climate change, such as ocean acidification and changes in temperature, salinity, or light [2–3]. In addition to their ecological significance, coral reefs constitute an invaluable economic resource, with plethora of natural products of biotechnological and pharmaceutical interest having been isolated from reef organisms, featuring anti-cancer, anti-microbial, anti-viral and anti-inflammatory properties [4–8]. Consequently, establishing in-vitro cultivation systems that will allow the identification of the factors that determine the successful settlement and development of coral planulae-larvae is of prime importance to the ongoing world-wide efforts for reef conservation [9–10].

One of the key aspects in coral life cycle and development is that of the interaction with their associated microorganisms. The coral holobiont is a complex symbiotic system that encompasses symbiotic dinoflagellate algae, bacteria, fungi, viruses, and other protists [11–13]. However, while the symbiotic relationship between the endosymbiotic dinoflagellate algae (zooxanthellae) and their coral host has been studied in depth [11, 14–16], the complex mechanisms controlling coral-bacteria interactions are still poorly understood. In particular, the specificity of the coral-bacteria association remains unclear, despite numerous studies having indicated a crucial role of bacteria for the well-being of corals [17–18]. High-throughput sequencing techniques have revealed the high bacterial diversity harboured by adult corals, with an increasing evidence of the existence of a species-specific "core microbiome" that seems to be regulated by the host through different mechanisms [19–22]. The coral probiotic theory [23], has led to the term "Beneficial Microorganisms for Corals" (BMCs) being coined, aimed at defining the specific bacterial symbionts that may promote coral health, and thus be of potential use as environmental probiotics to reverse dysbiosis [13]. The potential functions of this beneficial microbiota include carbon, nitrogen and sulphur cycling [24], as well as the production of anti-microbial compounds [25], thereby facilitating pathogen control.

Since the settlement and metamorphosis of coral planula-larvae are known to be governed by environmental cues, several studies have examined the role of bacteria in these processes, mainly for stony coral species. The best-known source of chemical morphogens for coral planulae are the crustaceous coralline algae (CCA) [26–28]. However, the nature of the biochemical cues present in CCA is not clear and may be partially derived from the presence of epiphytic bacteria [29]. Bacterial biofilms also produce cues for the settlement process that can be recognised by planulae [29–32], as previously reported for various marine invertebrates [33–34]. Indeed, the addition of antibiotics has been shown to block settlement and development in several Octocorallia species [35]. The nature of the biochemical and/or physical cues present in microbial biofilms remains elusive [36–37]. In many cases the percentage of settlement induced by monospecific bacterial strains was lower than that resulting from natural, multispecies films [32–33]. It is however possible that specific types of bacteria may be responsible for facilitating settlement and metamorphosis [36–38]. Members of the Roseobacter clade, a group frequently associated with stony corals [39], have been found to be constantly present in their planula-larvae [40–42], constituting up to 70% of the SSU rDNA sequences obtained [43]. Bacteria belonging to the genera *Alteromonas*, *Shewanella* and *Marinobacter* have also been associated with planulae or coral gametes in stony corals [41–42]. In other cases, members of the genera *Burkholderia*, *Pseudomonas*, *Acinetobacter*, *Ralstonia* and *Bacillus* have been reported to be transmitted vertically to their gametes [44]. A number of publications have also focused on the mode of transmission of the potentially beneficial bacteria to planulae [40, 42]. Several studies have demonstrated that, independent of the reproductive strategy of the corals,

specific, potentially beneficial bacteria are transferred vertically to the next generation during gamete or planulae release [41–42, 44]. In other cases, the bacterial populations seem to become established post spawning, according to the water bacterial community structure [43–45]. However, these studies, that indicate the importance of the presence of core bacteria genus, have been carried out with stony corals, with little information being available on the role of bacteria on Octocorallia settlement and metamorphosis [46], despite the importance of this group for the structure and trophic dynamics of coral reefs [47]. Moreover, most studies of this type are observational, with samples being obtained from the nature, without evaluating the performance of the analysed larvae for long periods.

Preliminary studies have indicated the relevance of bacteria for the settlement and development of zooxanthellate and azooxanthellate planulae of different octocoral species, since both processes are halted in the presence of antibiotics [35]. The overall goal of the current study was thus to evaluate the relevance of the presence of bacteria as BMCs for the settlement and development of octocoral planulae. Most studies to date have used next-generation high-throughput sequencing techniques in order to identify core microbiome species in the coral larvae or gametes, without examining the development of the larvae [41–45]. Here, however, we employed an experimental approach by cultivating the planulae in seawater filtered through different filter pore-sizes and/or autoclaved to limit the presence of bacteria, and consequently evaluate their effect on the development of the planulae. The effect of the addition of specific native bacteria to the seawater was also examined. Experiments were performed with planulae of the Red Sea octocoral *Rhytisma fulvum fulvum* (formerly *Parerythropodium fulvum fulvum*) [28, 35, 47]. *R. f. fulvum* is a zooxanthellate species, being one of the first species to recolonise areas where corals have died or damaged areas of reefs. This species is a surface-brooder: fertilized eggs cleave on the surface of the female colonies while entangled in a mucoid suspension [35, 47]. Our findings confirmed that bacteria acquired during the brooding period seem to be sufficient to support planulae development since mature larvae development could be achieved in sterilized seawater in plastic cell culture plates without the addition of natural coral substrata. Nevertheless, the results indicate the presence of bacteria in the 0.45 µm FSW that were beneficial for planulae metamorphosis. This culture system constitutes a valuable experimental model for further studies regarding changes in the planulae-associated microbiome during growth.

## Materials and methods

### Planula larvae collection

Planula larvae of the octocoral *Rhytisma fulvum fulvum* (Forskål 1975) were collected by scuba diving at 4–6 m depth in Eilat (northern Red Sea, Israel, 29°30′N, 034°55′E) during three different spawning events: July-2014, July-2015 and July-2016. The collection of animals complied with a permit issued by the Israel Nature and National Parks Protection Authority. The planulae were collected into zip-lock bags with a plastic transfer pipette. In the laboratory, they were washed several times with seawater passed through 0.45 µm pore filter before being distributed into the culture units. The experiments were initiated within 24 hours of planulae collection. The work was performed in the Interuniversity Institute for Marine Sciences (IUI, Eilat, Red Sea, Israel 29°30′N; 34°56′E).

### Culture water preparation

Ten litres of surface seawater were collected from the same location as that where larvae were harvested. Seawater was sequentially filtered with a 1.2 µm, GF/C or 2.7 µm, GF/D filter (Whatman™, Maidstone, UK)], and/or 0.45 µm and 0.2 µm cellulose nitrate filters (Sartorius,

Stedim Biotech GmbH). All processes were carried out using autoclaved Erlenmeyer glass flasks. No cultivable bacteria could be observed in the 0.22 μm FSW. The concentration of cultivable bacterial in the 1.2 μm FSW was $1.95 \pm 0.15 * 10^5$ CFU/mL in Marine Agar (MA, Difco) and $2.27 \pm 0.03 * 10^5$ CFU/mL in Tryptone Soy Agar (Difco) adjusted to 1% NaCl (TSA-1). The number of cultivable bacteria in the 0.45 μm FSW was $3.4 * 10^4$ CFU/mL independently of the culture medium used. CFUs were measured by re-suspending the bacteria retained in the filters in sterilized seawater. Direct enumeration of CFUs in the water yielded high variability and lower counts, indicating low homogeneity of the sample and/or the presence of bacteria attached to particulate material.

## Native bacterial isolation and identification

Cellulose nitrate filters (0.45 μm and 0.2 μm) were re-suspended in autoclaved seawater and aliquots were plated on MA and TSA-1 and incubated for 7–10 days at 25˚C in the dark. In order to examine the effect of these isolates on coral larvae settlement, selected isolates were incubated on 10 mL Marine Broth or TSB-1 in 25 mL Erlenmeyer flasks in a shaker for 24–48 hours at 25˚C. Bacterial cells were harvested by centrifugation and washed 3 times with autoclaved seawater in order to remove any organic trace from the culture medium. Bacteria were added to the planulae cultures at a final concentration of $10^3$ CFU/mL with every water exchange. Cell density was estimated as optical density at 600 nm, after generating CFU/OD calibrating curves for each species.

An additional experiment was carried out with larvae obtained in 2015 in which the wells were pre-conditioned with bacterial biofilm before the planulae were added [48]. Different volumes (50, 100 and 200 μL) of bacterial suspensions is sterilized sweater were obtained as explained above and were inoculated in in the wells containing 5 mL of sterilizes sea water. Plates were incubated 25˚C for 48 hours in order to allow the attachment of the bacteria. Water was gently removed from the wells and refilled with 0.45 μm FSW for the cultivation of the planulae.

The identification of strains was based on 16S rRNA gene sequencing. Bacterial DNA was extracted using a Wizard DNA Purification Kit (*Promega*, Southampton, UK) as per manufacturer´s instructions. The amplification of 16S rDNA gene sequences were done by polymerase chain reaction (PCR) with the primers 96bfm (5′-GAGTTTGATYHTGGCTCAG-3′) and 1152uR (5′-ACGGHTACCTTGTTACGACTT-3′) [49], the GoTaq DNA Polymerase (*Promega*, Southampton, UK). PCR were carried out under the following standard conditions: initial step of 96˚C for 2 min followed by 35 cycles of 95˚C for 1 min, 53˚C for 30 s and 72˚C for 2 min [50]. The 16S rRNA sequences were identified using the web-based tool EzTaxon (https://www.ezbiocloud.net/).

## Larvae cultivation

Planula cultures were carried out in untreated 6-well tissue polystyrene culture plates (Jet Biofil®) containing 10 coral larvae per replicate (n = 3) with a final larvae concentration, 1 larvae/mL. All bioassays with coral larvae were carried out at 24±1˚C in an incubator. In some experiments the cultures were maintained under a 12:12 light:dark cycle. 50% of the volume of the cultures was exchanged on alternate days. The same batch of FSW or ASW was used for the whole duration of each experiment. No evaporation was observed in the cultures throughout the experiments. Some experiments were also carried out in sterilized 60 mm glass Petri dishes with 10 mL of sea water and a concentration of 10 planula-larvae per mL.

Planulae settlement and development was monitored daily or every 2 days using a stereoscopic microscope. Development was classified into four different categories: pre-metamorphosed; early metamorphosed and advanced metamorphosed, subdivided according to polyps

**Fig 1. Morphological classification of *R. f. fulvum* planulae early developmental stages.** The planulae were classified into four main developmental stages: pre-metamorphosed, early metamorphosed, advanced metamorphosed polyps with short tentacles and advanced ones with long tentacles.

with short or long tentacles (Fig 1). The pre-metamorphosed larvae comprised of coccoid or swimming stage larvae. The early metamorphosed comprised under-developed polyps, usually attached and with only the paddle-disc visible. the advanced metamorphosed stages are fully developed polyps with feather-like tentacles (short or long).

## Statistical analysis and data treatment

The statistical analysis was performed in R software (version 3.3.1) using "coin" package [51] and function Wilcox_test. Since the data did not fulfill the conditions of normality and homoscedasticity, and could not be improved by transformation, they were analyzed using the non-parametric Wilcoxon-Mann-Whitney test with exact distribution of the statistic due to the limited data available. To determine the effect of water exchange and autoclaved water, we took into account the 12 values obtained from the four different types of water considered in the study, in order to obtain at least a minimum sample size. The remainder of the analyses were computed with six values, three for each of the conditions considered here.

## Results

### The influence of culture conditions and water treatment on larvae survival and development

The response of *R. f. fulvum* planulae to the effect of filtering the seawater (FSW) through 1.2, 0.45 and 0.22 μm pore filters or autoclaving it (ASW) was assessed in polystyrene 6-well tissue culture plates within 24 hours of being harvested from the environment. Despite previous reports had indicated that planulae settlement can occur within 24–48 hours following transfer to suitable conditions [29, 35], no settlement signs could be observed in the first 48 hours in an initial experiment carried out with planulae harvested in June 2014 (S1 Fig) or in any of the subsequent experiments carried out in 2015 and 2016, independently of the water treatment applied. The 2014 cultures were maintained without water change and all planulae were dead by day 12 of incubation (S1 Fig). Nevertheless, the results of these preliminary experiments indicated that larvae maintained under dark conditions presented a better survival trend than those maintained under light:dark (L:D) cycles, since all larvae were alive by day 6 in the dark, regardles the water treatment, while 50% of the larvae were aleady dead in some of the treatments in L:D cycles (S1 Fig. Also, the 0.45 μm FSW presented a better survival trend since the number of live larvae on day 10 was significantly higher that with the other water treatments (Wilcoxon-Mann-Whitney Test, p<0.01, S1 Fig).

The same experiment was repeated with planulae harvested in July 2015. The cultures were maintained for 32 days by exchanging of 50% of the culture water under light:dark diurnal cycles. Under such conditions the lowest final survival rate (70%) was obtained with ASW,

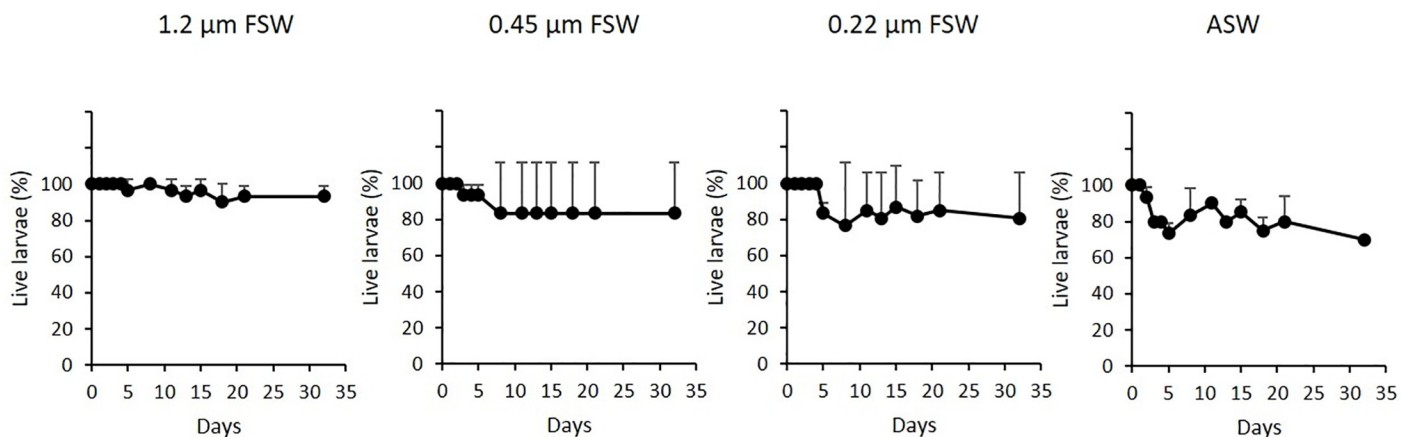

**Fig 2. Average number of live planulae of *Rhytisma fulvum fulvum* over time in cultures maintained with different levels of filtered seawater (FSW) and autoclaved seawater (ASW).** Cultures were maintained with water exchange of 50% of the volume of the cultures on alternative and under light:dark cycles (12:12 hours). The experiment was performed with planulae harvested in 2015. Data are shown as means ± S.D, (n = 3, 10 planulae per replicate).

while 1.2 μm FSW produced the highest survival values (93.3% ± 5.8, Wilcoxon-Mann-Whitney Test, p<0.01, Fig 2). No difference in survival (80%) was noted between 0.22 and 0.45 μm FSW after 32 days of incubation (Wilcoxon-Mann-Whitney Test, p>0.05). No improvement in survival or attachment was observed when the same cultures were carried out in glass Petri dishes at the same planulae density (S2 Fig). Regarding metamorphosis of the planulae, despite the absence of a CCA substratum, a high number of metamorphosed planulae was observed in the cultures after 32 days of incubation, with values higher than 50% in all treatments (Fig 3). In this experiment all planulae from the same treatment were classified together and the advanced metamorphosed stage was differentiated between short and long-tentacle stages as in later experiments. The best results were obtained with 0.22 μm FSW, achieving a 72.4% rate of metamorphosed planulae, mainly in advanced stage, in comparison to 55% obtained with 0.45 μm FSW and 57.1% with ASW (Fig 3). Metamorphosis of planulae maintained in 1.2 μm FSW was even lower, reaching only 50% (Fig 3). Survival rates higher than 20% were obtained on day 32 in the same experiment for larvae that were maintained without water exchange (S3, Fig) for comparison with the 2014 experiments, when all larvae were dead by day 12 (S1, Fig) thus indicating strong annual variations in planulae performance. Nevertheless, poor development was observed in stagnant cultures, with none entering the advanced metamorphic stage.

On the basis of the higher survival values obtained in the 2014 experiment carried out under stagnant conditions during the first days of cultivation when the planulae were maintained in the dark in comparison with those maintained under L:D diurnal cycles (S1 Fig), the effect of maintaining the cultures in the dark or under L:D cycles and with three different water treatments (2.7, 0.45 or 0.22 μm FSW) was determined with a batch of planulae obtained in June 2016 (Fig 4). The pore-size of the largest filter was increased to 2.7 μm in comparison with previous experiments (1.2 μm) in order to allow the presence of small phytoplankton cells in the culture water that could be beneficial for planulae development. In this experiment, 100% survival was observed in all treatments on day 35 of incubation regardless the light regime. Notably, significant differences in the evolution of the percentage of metamorphosed lavae were recorded in relation to light conditions (Fig 4). Darkness seemed to initially stimulate metamorphosis, since the number of metamorphosed planulae in cultures maintained under such conditions on day 20 was twice the number obtained in normal light-dark maintained cultures (Wilcoxon-Mann-Whitney test, p<0.01, Fig 4), but clearly decreased thereafter, indicating a regression of the initial attachment states. On the contrary, cultures

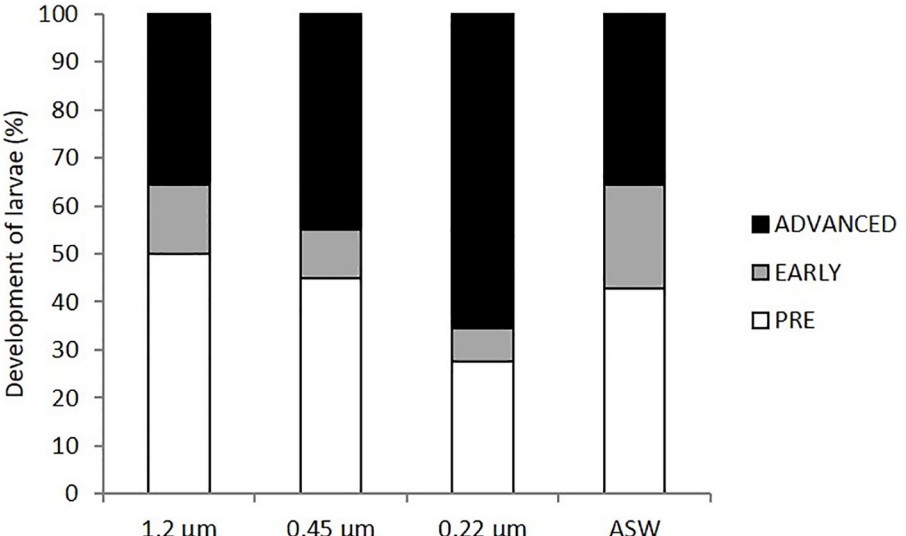

**Fig 3. Development of *Rhytisma fulvum fulvum* planulae on day 32 of incubation.** Planulae were maintained with different levels of filtered seawater (FSW) and autoclaved seawater (ASW) with water change on alternative days. Results correspond to the same experiment shown in Fig 1. Planulae development was classified into three stages: pre-metamorphosed (open bars); early metamorphosis (grey bars); and advanced metamorphosed (dark bars). Cultures were maintained under light:dark cycle (12:12 hours). The advanced metamorphosis stage corresponds to the Advanced-short plus Advanced-long stages described in Materials and Methods. Percentages were calculated on the basis of the analysis of all survived larvae.

maintained under light:dark conditions presented a continuous increase in the number of metamorphosed planulae, indicating that light promotes the achievement of full metamorphosis. Nevertheless, no statistically differences in the total metamorphosed larvae were observed between dark and light-dark maintained cultures by the end of the experiment (Wilcoxon-Mann-Whitney test, p = 0.8). Metamorphosis was accelerated after day 20 in the 0.45 and 0.22 FSW cultures maintained under a light:dark cycles, achieving 76.7% ± 20.8 and 73.3% ± 28.9 metamorphosed larvae on day 35 respectively, with no significant differences among them (Wilcoxon-Mann-Whitney test, p = 1). These values were higher than those obtained in the dark maintained cultures, 63.3% ± 37.9 for 0.45 FSW and 53.3%± 15.3 for 0.22 FSW but the differences were not statistically significant (Wilcoxon-Mann-Whitney test, p>0.6). Only in the 2.7 μm FSW did the dark-maintained cultures display a better rate of total metamorphosed larvae on day 35, (66.6% ± 15.3) than in the light-dark cycles (43.3% ± 10, Fig 4), although differences were not statistically significant (Wilcoxon-Mann-Whitney test, p = 0.4). When the level of development of the metamorphosed larvae is considered, the best metamorphosed cultures were those maintained in 0.45 FSW in a light:dark cycle, with 63.3% of the planulae reaching the advanced, long-tentacle metamorphic stage at day 35, compared to 3.3% of those in the dark (Fig 5), although none of these differences were statistically significant) (Wilcoxon-Mann-Whitney test, p>0.1). Despite the similarity in the percentage of total metamorphosis, the percentage of primary polyps with long tentacles under a light:dark cycles on day 35 was also significantly higher in the 0.45 FSW treatment than in the 0.22 FSW (63.3% vs. 26.7%, respectively) (Fig 5). When examined under a fluorescence microscope, no zooxanthellae could be observed in any of the cultures after 35 days, indicating that the symbiotic algae are not required, at least for the initial metamorphic stages.

The three types of FSW were also tested with or without autoclaving under light:dark (12:12 h) cycles, in order to determine whether the observed differences were derived from the

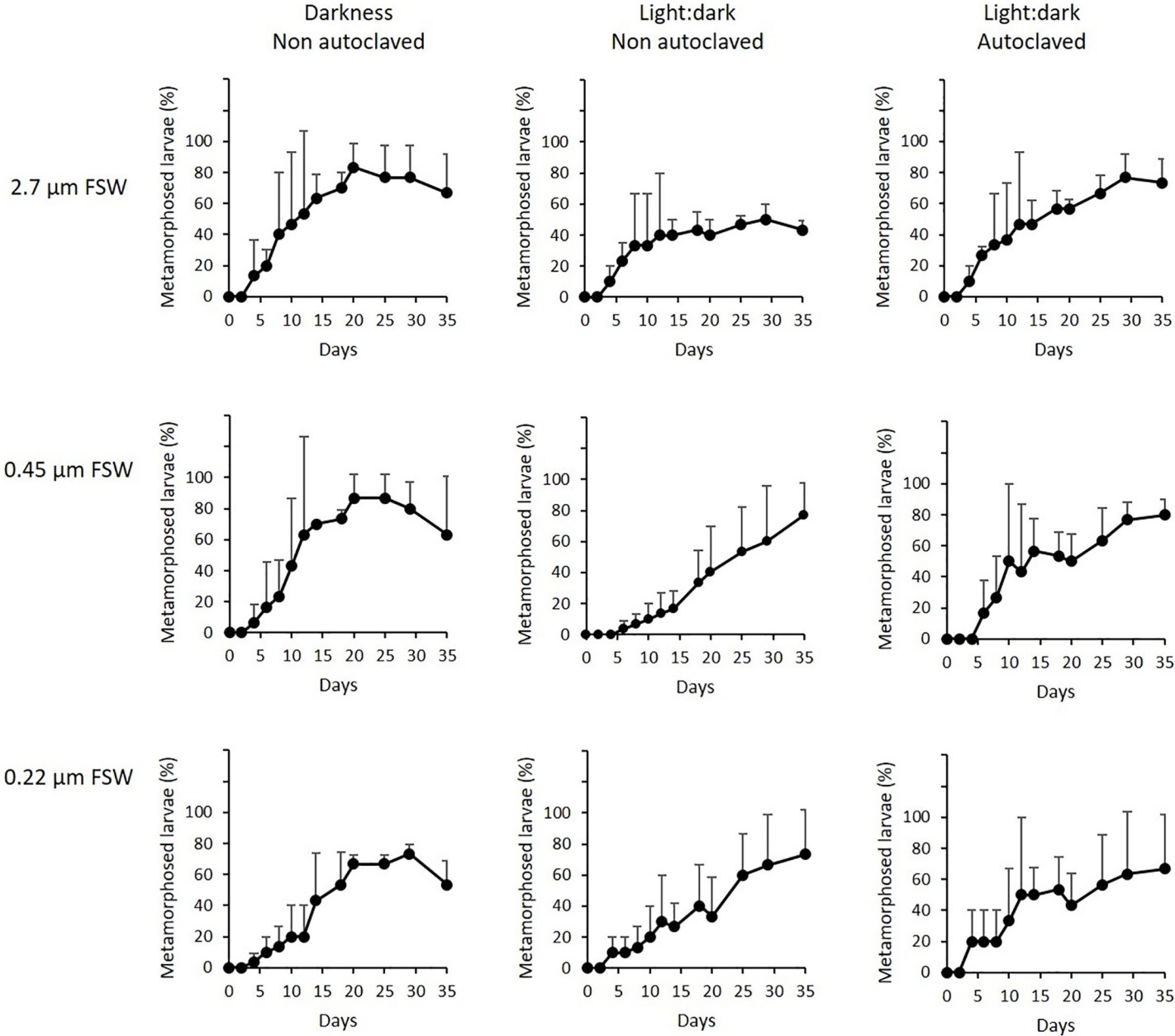

**Fig 4. Number of metamorphosed *Rhytisma. fulvum fulvum* planulae in cultures maintained with different pore-size filtered seawater (FSW).** Seawater was filtered through 2.7 µm, 0.45 µm, 0.22 µm. For the cultures maintained in a light:dark cycle the effect of autoclaving the different FSW was also determined. Survival was 100% in all treatments on day 35. Half of the culture water was exchanged on alternative days. The experiment was performed with planulae harvested in 2016. Data are shown as means ± S.D. (n = 3, 10 planulae per replicate).

presence of live bacteria (Fig 4). Autoclaving the different FSW homogenized the development of the larvae among the different treatments, achieving values of total metamorphosis of 73.3% ±15.3 for 2.7 µm FSW, 80% ± 10 for 0.45 µm FSW and 66.7% ± 35.1 for 0.22 µm after 35 days of incubation with no statistically significant differences among them (Wilcoxon-Mann-Whitney test, p = 0.4, Fig 4). When comparing between the autoclaved FSW and the non-autoclaved, the autoclaving of water clearly improved metamorphosis in the 2.7 µm FSW, increasing from 43.3% ± 0.58 to 73.3% ± 15.3, although this difference was not statistically

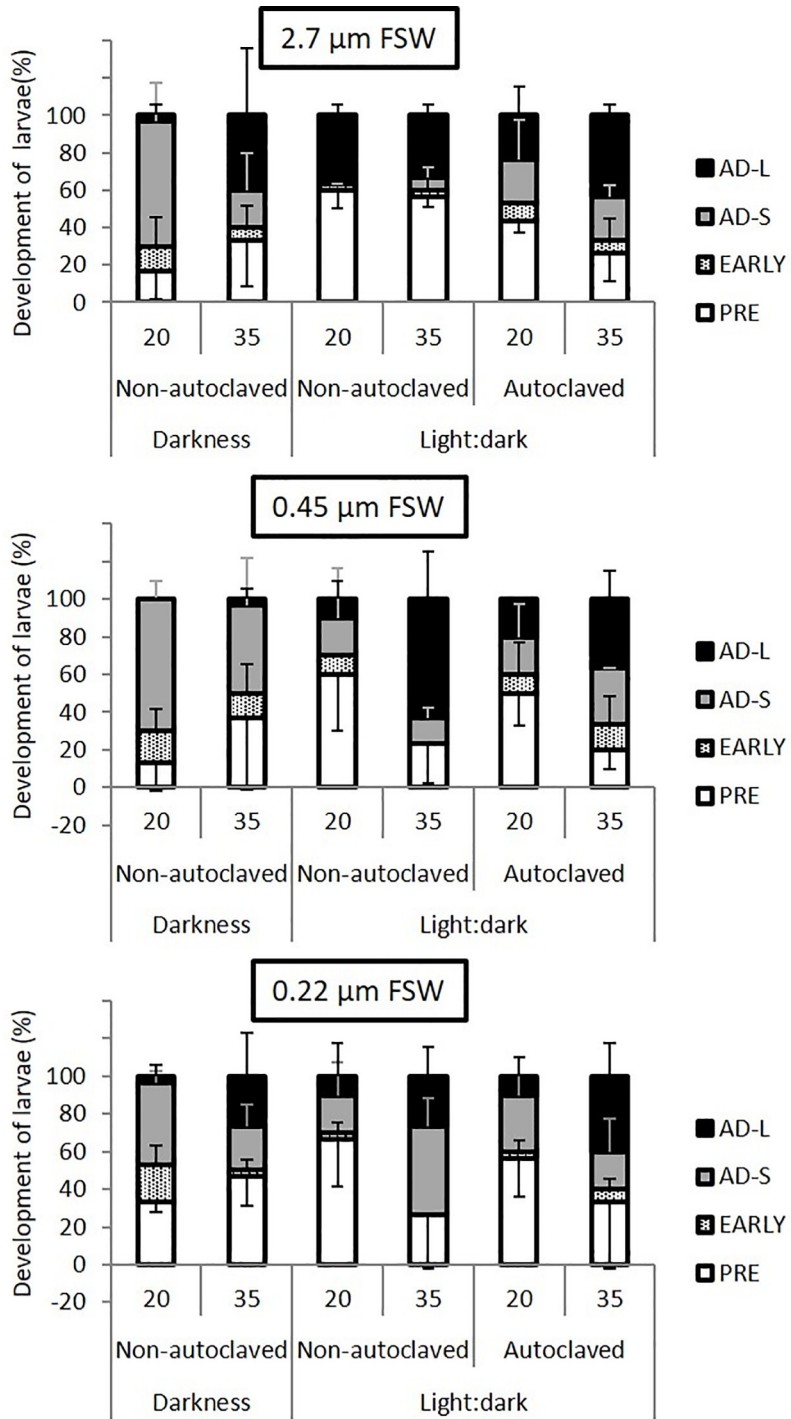

**Fig 5. Development stages achieved by *Rhytisma fulvum fulvum* on days 20 and 35 of culture under different water treatments and light conditions.** Cultures are the same shown in Fig 3. Data are presented as percentage of total (n = 3, 10 larvae per culture unit, no dead larvae observed during the period). Planulae were classified into pre-metamorphosed (open bars), early metamorphosed (striped bars), advanced metamorphosed with short tentacles (AD-S, grey bars) and advanced metamorphosed with long tentacles (AD-L, black bars). Data are shown as means ± S. D. (n = 3, 10 planulae per replicate).

significant (Wilcoxon-Mann-Whitney test, p = 0.1, Fig 4). Regarding the developmental stages achieved in the different treatments (Fig 5), the number of primary polyps with long tentacles at the end of the experiment in the autoclaved seawater treatments were in the range 36.7–43.3%, regardless of the filter used in the pre-treatment, with no statistical difference among them (Wilcoxon-Mann-Whitney test, p = 0.3, Fig 5). Although autoclaving the different FSW did not produce statistically significant differences in comparison with the non-autoclaved counterparts (Wilcoxon-Mann-Whitney test, p = 0.3, Fig 5), the number of primary polyps with long tentacles increased from 26.7% ± 15.3 to 40%± 17.3 the autoclaving the 0.22 μm FSW (Fig 5). In contrast, autoclaving the 0.45 μm FSW seem to negatively affected the planulae, since the number of primary polyps with long tentacles decreased from 63.3% ± 25.2 to 36.7% ± 15.3, although featuring a similar rate of total metamorphosis (Fig 5). These results indicate that the beneficial effects of the 0.45 μm FSW may be derived from the presence of live bacteria in the water.

## Effect of native bacteria on coral larvae survival

Following our findings indicating the possible presence of beneficial bacteria for the induction of metamorphosis in the 0.45 μm FSW, we isolated bacteria from the 0.45 and 0.22 filters, representative of the bacteria in the 1.2 μm and 0.45 μm FSW respectively and examined the effect of the addition of $10^3$ CFUs mL$^{-1}$ on the planulae. Several Gram-negative isolates, belonging to the Alpha-proteobacteria, including two members of the Roseobacter clade *(Ruegeria mobilis* and *Mameliella atlantica)* and to the *Gamma-proteobacteria (genera Alcanivorax, Marinobacter, Alteromonas* and *Vibrio)* (S1 Table) and two Gram-positive isolates belonging to the genera *Kokuria* (isolate A2) and *Planomicrobium* (isolate 23) were tested. Several experiments carried out in 2014 and 2015 in which the bacteria were added to the larvae cultures and maintained without water exchange (S4 Fig. and S5 Fig.) indicated that the addition of specific bacteria could greatly improve the survival under such conditions, probably by helping in maintaining the water quality within acceptable physiological values. The strains *Alteromonas macleodii* (P9) and *Ruegeria mobilis* (1) allowed achieving survival values in the range of 80–90% in comparison with a 17% of survived larvae in the control cultures without water exchange after 32 days in the 2015 experiment (S5 Fig) (Wilcoxon-Mann-Whitney, p<0.01), while in the 2014 experiment 76% of the larvae were alive after 12 days, while all the larvae in the control cultures were dead (S4 Fig). In an additional experiment in which the effect of bacteria was tested with our without water exchange for the 2015 larvae, *A. macleodii* P9 positively affected bot, larval survival and development in the cultures maintained without water exchange, allowing to achieve a 93% of early metamorphosed larvae in comparison with th4 40% in the control cultures (S6 Fig.). This beneficial effect disappeared with water exchange (S6 Fig). Since the presence of bacterial biofilm has been described as necessary for planulae attachment and metamorphic development, we also attempted to pre-condition the cell culture units [48] with biofilms of two native *Alteromonas macleodii* strains (strains M1 and P14) and several marine strains that are strong biofilm formers: *Pseudoalteromonas flavipulchra*, *Pseudoalteromonas maricaloris* [29,30, 36, 52, 53], *Vibrio tubiashi* and *Vibrio aestuarianus* [54]. None of the tested strains induced planulae attachment in comparison with control cultures.

The effect of the addition of native bacteria was tested on planulae obtained in 2016 with water exchange performed on alternative days. Bacteria were added with every water exchange. In this experiment 0.22 μm FSW was used in order to reduce potential noise generated by the continuous addition of bacteria with the water. No mortality was recorded in any of the treatments, including the control that was maintained with bacteria-free water, indicating that the

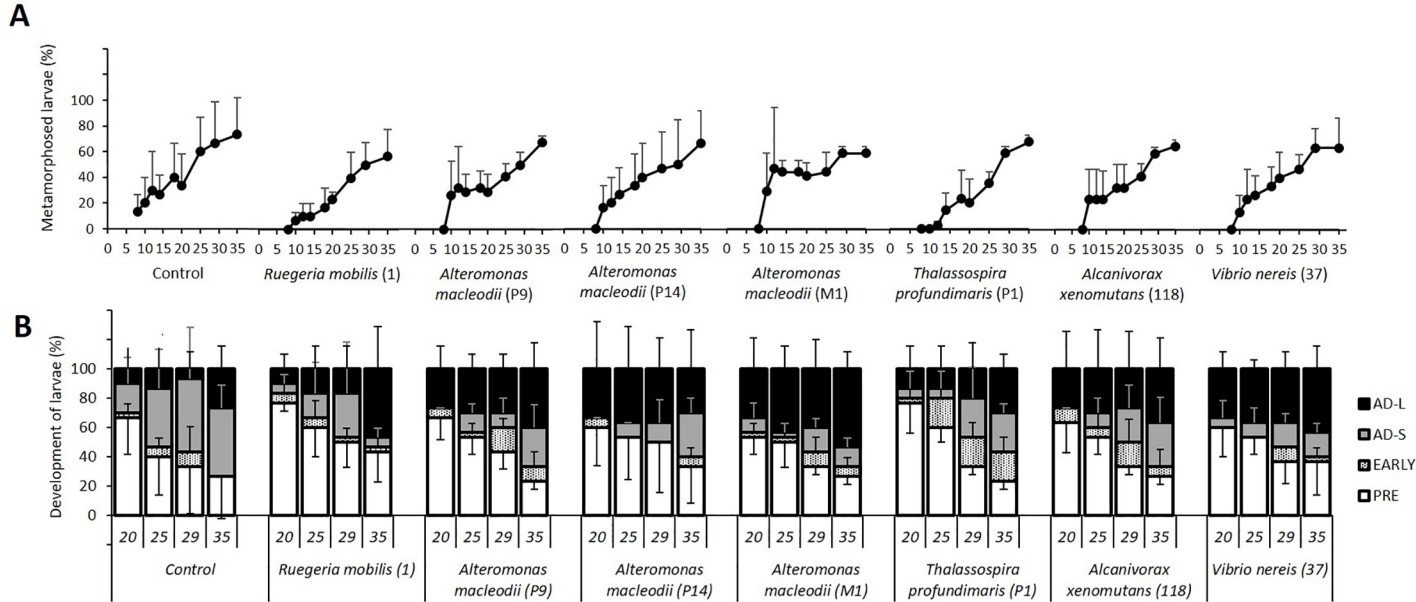

**Fig 6. The effect of native bacteria (10³ CFU mL⁻¹) on the metamorphosis of *Rhytisma fulvum fulvum* coral planulae.** Planulae harvested in 2016 were maintained with water exchange on alternative days and under a 12L:12D h photoperiod condition. All planulae were alive after 35 days of incubation. A. Total metamorphosed planulae on days 8, 10, 12, 14, 18, 20, 25, 29 and 35 of the experiment. B. Developmental stages on days 20, 25, 29 and 35. Planulae were classified into four stages: pre-metamorphosis (open bars), early metamorphosis (stippled bars); advanced metamorphosis with short tentacles (grey bars); and advanced metamorphosis with long tentacles (black bars). Data are presented as means ± S.D, n = 3, 10 individuals per replicate).

exogenous addition of bacteria is not essential for survival and metamorphosis, at least when the initial physiological and/or microbiological state of the larvae is adequate. The addition of bacteria did not significantly change the number of fully metamorphosed planulae which reached values between 56.7 and 76.7% on day 35 (Wilcoxon-Mann Whitney test p>0.1, Fig 6A). As observed in the 2015 experiment (Fig 4), a steep increase in development was observed from days 18–20 (Fig 6A). Only non-statistically significant differences were found in the metamorphosis stage of planulae in the presence of bacteria (Fig 6B). *A. macleodii* M1 produced a higher number of primary polyps with long tentacles (5.33±1.15) than the control (2.67±1.53) on day 35, while producing the same number of metamorphosed planulae (Fig 6B). The Roseobacter clade member *R. mobilis* 1 also led to a higher number of long-tentacle primary polyps (4.67±2.89) but the total number of attached larvae was lower with this strain in comparison with the bacteria-free control. Minor beneficial effects were noted with the addition of *V. nereis* 37 (long-tentacle: 4.33±1.43) and *A. macleodii* P9 (long-tentacle: 4±1.73).

## Discussion

Understanding the contribution of the biological and physico-chemical factors that control coral-planulae settlement and metamorphosis is critical for identifying the main elements affecting coral population dynamics, and for future modelling of the effects of climate change and human disturbances on coral reefs. Increasing evidence points to a crucial role of bacterial biofilms in the recruitment of marine invertebrate-larvae [38], including coral planulae [29, 32, 52]. The relative importance of environmentally-acquired versus vertically-transferred bacterial populations for larval settlement and development is not yet fully understood. Most studies are focused on metagenomic analysis of stony coral colonies and larvae [41–45], or testing the effect of particular bacteria on very early development [29, 32, 46, 52]. It is also necessary

to establish simple and reliable culture protocols that allow the study of the changes in the coral larvae microbiome during different development phases under controlled conditions. The current study thus sought to elucidate the role of bacteria in the metamorphosis of octo-coral *R. f. fulvum* planulae employing different types of filtered and autoclaved seawater and through the addition of native bacteria. To date most studies seeking to identify the cues controlling larval settlement have used short incubation periods (from 24–48 hours to a few days) [29, 52]. Consequently, little information is available regarding the long-term effect of bacteria and incubation conditions on larval settlement and metamorphosis.

In contrast to previous results that reported attachment of *R. f. fulvum* within 24–48 hours in the presence of natural reef substrata [28, 35], the first attached larvae in the cell culture wells were observed on day 4–6, independently of the water treatment and light regime applied (Fig 4). Indeed, in the presence of crustose coralline algae, *R. f. fulvum* planulae settle exclusively on this substratum, avoiding glass or plastic surfaces, with a clear preference for live corals instead of bleached fragments [28]. In our case, although the absence of coral substrate may have been the cause of the delayed settlement of the larvae, the presence of organic coral substrata or the associated bacteria does not seem to be necessary to achieve a good development rate of the planulae. Values of 60–85% of metamorphosed larvae were obtained in the plastic cell culture wells after 20 days of incubation in the dark, and 40–50% in light-dark diurnal cycles even with autoclaved water (Figs 4 and 5). Notably, the value of metamorphosed *R. f. fulvum* planulae reported in larger aquaria (5 litres, 100 planulae per litre) in the presence of live natural reef substratum was 58% after the same incubation period [35]. A good metamorphic rate was also obtained in the plastic cell-culture plates, with long tentacle primary polyps being observed on day 20 of incubation even with 0.22 FWS (Fig 6), whereas similar results were obtained only after two months of incubation in larger culture systems [36]. The ecological relevance of bacterial cues for larval attachment has been questioned recently, while the importance of CCA-derived cues seems to be confirmed [27]. As for *R. f. fulvum* planulae [28], the presence of CCA biofilm has been reported to clearly stimulate larval settlement in the planulae of the octocoral *Sinularia polydactyla*, that could not be achieved on a plastic surface [46]. In the current study, successful attachment and metamorphosis were achieved even in the absence of calcareous substrates, and no differences were found between plastic and glass surfaces, which suggests that settlement cues should be species-specific or strongly dependent on experimental conditions. The number of planulae that can be harvested from the environment is often a limiting factor in coral research. Therefore, these results demonstrate that, at least for *R. f. fulvum*, cell culture plates can be used as a simple, convenient experimental method that allows testing a large number of conditions with a limited number of planulae, as far as water conditions are maintained within appropriate values.

Previous reports demonstrated that the addition of antibiotics completely halted the development of *R. f. fulvum* development, indicating that bacteria are necessary for the metamorphic process [35]. Our results strongly support the idea that the bacteria acquired during brooding are sufficient to support larval settlement and development in *R. f. fulvum*, in the view of the acceptable metamorphic rates obtained in the cultures maintained with the bacteria-free 0.22 FSA and ASW (Figs 4 and 5). In the case of *R. f. fulvum*, the very specific antimicrobial activity found in larvae and adults may be related to the selection of certain beneficial bacteria to be closely associated with their coral host, as well as excluding potentially pathogenic bacteria [55]. Indeed, bacterial symbionts seem to be transferred vertically in brooding corals [41], whereas broadcast-spawning corals seem to acquire their associated bacterial communities post-settlement [43, 45]. However, adult corals may release specific genera of bacteria during the spawning events in order to benefit the fitness of their sexual progeny [42]. In any case, a higher percentage of metamorphosed planulae with long tentacles was obtained with

the 0.45 μm FSW in comparison with the 0.22 μm FSW (Fig 5). Although this beneficial effect should be confirmed in further experiments, in the view of the observed yearly differences in larvae performance, and with a higher number of replicates, in order to allow a more reliable statistical analysis, the beneficial effect of the 0.45 μm FSW depended on the presence of live bacteria in the seawater, since autoclaving the 0.45 μm FSW negatively affected larval development (Fig 5). Such a beneficial effect may be related to different positive effects, such as competitive interactions in the bacterial population that results in the exclusion of pathogens, a direct probiotic effect or the stabilisation of the water chemistry. The analysis of the evolution of the species composition of the coral-associated microbiome under different culture conditions with metagenomics techniques will surely add valuable information for assessing the role of bacteria on coral development.

A comparison of the results obtained with and without light revealed a clear two-stage pattern of the planulae metamorphosis in *R. f. fulvum*. Settlement was initially accelerated in the dark, but the cultures underwent a reversion in development after day 20, regardless the type of water used for maintaining the cultures (Fig 4). This stimulation of development in the dark can be related to a preference of the planulae to settle on the dark side of natural substratum [28]. In contrast, settlement and metamorphosis were clearly seen to accelerate after day 20 in the cultures maintained with 0.22 μm FSW in the light (Figs 4 and 6). A less marked trend was also observed in cultures maintained with 0.45 μm FSW in the light, starting on day 16 (Fig 4). Autoclaving the water seem to produce a more continuous trend in the rate of total metamorphosed planulae (Fig 4). Although light can be used as a positive cue for larvae development in several cnidarians species [28] we cannot exclude that light acts through the stimulation of the microbial component of the holobiont. *R. f. fulvum* planulae are devoid of the photosynthetic algal symbiont upon release [35]. Similarly, in our cultures, fluorescence microscopy did not reveal the presence of any algal cells in the metamorphosed planulae, suggesting that despite *R. f. fulvum* being a zooxanthellate species, establishment of algal symbiosis is not required for the initial metamorphosis stages. We cannot completely disregard the idea that the beneficial effect of light seen after day 16–20 on the larval development may be due to the presence of phototrophic bacteria in the holobiont. Indeed, the Cellvibrionales BD1-7 (previously Alteromonadales), a phototrophic, proteorhodopsin-containing group, is among the most abundant bacterial taxa associated to Mediterranean octocorals [22]. In contrast, the initial higher metamorphic rate of planulae maintained in the dark could be related to their preference for micro-habitats of low light intensity [28, 56]. The use of 2.7 FSW was clearly detrimental to larval development only under light conditions, but development improved when the water was autoclaved (Fig 4), indicating the possible presence of photosynthetic microorganisms detrimental to larval development in the 2.7 FSW. Such microorganisms might either compete for some limiting nutrient or produce a toxic compound. The presence of algae has been previously described to reduce the survivorship and settlement success of planulae, indicating the complexity of microbial dynamics in coral development [57]. Further experiments are required in order to elucidate if the beneficial effect of light on larval development is derived from the presence of beneficial phototrophic bacteria.

Despite some specific bacteria had a beneficial effect on larval survival when the water quality of the cultures was not controlled (S4 Fig., S5 Fig. and S6 Fig.), this beneficial effect disappeared almost completely under favourable conditions. Nonetheless, the addition of several *Alteromonas* strains, *Vibrio nereis* and the member of the Roseobacter clade *Ruegeria mobilis* seemed to favour larval development. Although the addition of these strains to planulae maintained in 0.22 FSW led to only minor improvements in their metamorphosis (Fig 6), it should be noted that the bacteria were added at a concentration $10^3$ CFU mL$^{-1}$, whereas concentrations of $10^6$ CFU mL$^{-1}$ are commonly used in aquaculture probiotic experiments [58]. It is

nevertheless highly improbable that the planulae encounter such a high concentration of CFUs of a single species under natural conditions. Species belonging to the genus *Alteromonas* and the *Roseobacter* clade have been previously associated with beneficial effects and increased settlement in coral larvae. The genera *Alteromonas* and *Roseobacter* dominated the taxa of bacteria released by adult corals in the case of the broadcast-spawning coral *Acropora tenuis* and the brooding stony coral *Pocillopora damicornis* [42]. Roseobacter-clade associated bacteria are also consistently detected in specimens of planulae of the brooding scleractinian coral *Porites astreoides* [41]. *Pseudoalteromonas*, a genus of Gammaproteobacteria closely related to *Alteromonas*, has been frequently related to increased settlement rates in different stony corals [29, 52] and other invertebrates [36]. In addition, the specific biochemical cues that induce metamorphosis in coral planulae have been identified for this genus [30, 53]. It should be noted that not all the tested *Alteromonas* strains produced the same beneficial effects (Fig 5, S5 Fig. and S6 Fig). Differences in metamorphosis-inducing characteristics between strains of the same species have been reported previously [36] and may derive from differences in the production of biochemical cues. In the current study, the production of essential growth factors or biochemical cues by such specific bacteria does not appear to be a crucial contributing factor, in the view of the good survival and metamorphosis obtained when the bacteria-free ASW or 0.22 FSW were used for the water exchange (Fig 2, Fig 3 and Fig 6). It is possible that the addition of specific bacteria helps to maintain the chemical homeostasis of the medium by removing toxic excreted compounds, or that an anti-microbial compound is produced that could help to reduce the number of pathogenic bacteria [59]. Anti-microbial activity is common among *Alteromonas* and *Pseudoalteromonas* species [59, 60], which could also explain the highly beneficial effect of the addition of *Alteromonas* strains to the cultures maintained under non-controlled water quality conditions. Future experiments, with a higher number of replicates and in larger seawater volumes, are required in order to confirm this beneficial effect and to assess the possible mechanism involved, together with their effect on the planulae microbiome.

## Conclusions

The current results clearly indicate that bacteria acquired during the brooding period are sufficient to sustain settlement and development in the zooxanthellate planulae of the surface-brooding octocoral species *R. f. fulvum*. This confirms the hypothesis of a strong host-driven control of the bacterial component of the holobiont. Further studies are required in order to characterize the bacterial population accompanying the settlement and metamorphosis processes in octocoral planulae. The feasibility of obtaining advanced metamorphosed larvae in sterilized water provides an invaluable tool for studying the physiological role of these bacterial symbionts in the coral holobiont.

## Supporting information

**S1 Fig. Survival of planulae the octocoral *Rhytisma fulvum fulvum* maintained with different filtered seawater (FSW) and autoclaved sea water (ASW).**
(DOCX)

**S2 Fig. Survival of planulae the octocoral *Rhytisma fulvum fulvum* maintained in plastic and glass plates and different filtered sea water (FSW) and autoclaved sea water (ASW).**
(DOCX)

**S3 Fig. Survival of planulae the octocoral *Rhytisma fulvum fulvum* maintained without water exchange with different filtered sea water (FSW) and autoclaved sea**

water (ASW).
(DOCX)

**S4 Fig. Survival rates of *Rhytisma fulvum fulvum* planulae in the presence of different native bacteria in cultures without water exchange.**
(DOCX)

**S5 Fig. Survival rates of *Rhytisma fulvum fulvum* planulae in the presence of different native bacteria in cultures without water exchange.**
(DOCX)

**S6 Fig. Influence of four native bacteria on the survival and development of planula larvae of the octocoral *Rhytisma fulvum fulvum* in cultures with and without water exchange.**
(DOCX)

**S1 Table. Identification of native bacteria isolated from superficial seawater in the Red Sea coral reef seawater tested on the planula larvae cultures of *Rhytisma fulvum fulvum*.**
(DOCX)

## Acknowledgments

We thank Prof. Yehuda Benayahu's group from Tel Aviv University and the staff of the Inter-university Institute for Marine Sciences in Eilat (IUI) for their kind hospitality and facilities, in particular Viviana B. Farstey and Noa Eden. We also thank N. Paz for English editing.

## Author Contributions

**Conceptualization:** Yehuda Benayahu, Ana Otero.

**Data curation:** Isabel Freire, Andrea Muras.

**Funding acquisition:** Yehuda Benayahu, Ana Otero.

**Investigation:** Isabel Freire, Eldad Gutner-Hoch, Andrea Muras.

**Methodology:** Eldad Gutner-Hoch, Andrea Muras, Ana Otero.

**Project administration:** Ana Otero.

**Supervision:** Ana Otero.

**Visualization:** Isabel Freire.

**Writing – original draft:** Isabel Freire, Ana Otero.

**Writing – review & editing:** Eldad Gutner-Hoch, Yehuda Benayahu, Ana Otero.

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
