## [Decision Letter · Decision Letter 0]

26 Jul 2019

PONE-D-19-17240

The effect of bacteria on settlement and metamorphosis in planula-larvae of the octocoral Rhytisma fulvum fulvum

PLOS ONE

Dear Prof Otero,

Thank you for submitting your manuscript to PLOS ONE. After careful consideration, we feel that it has merit but does not fully meet PLOS ONE’s publication criteria as it currently stands. Therefore, we invite you to submit a revised version of the manuscript that addresses the points raised during the review process.

As you can see both reviewers ask for some minor changes and correction, so please modify your manuscript according to these comments and resubmit it

We would appreciate receiving your revised manuscript by Sep 09 2019 11:59PM. To enhance the reproducibility of your results, we recommend that if applicable you deposit your laboratory protocols in protocols.io, where a protocol can be assigned its own identifier (DOI) such that it can be cited independently in the future. For instructions see: http://journals.plos.org/plosone/s/submission-guidelines#loc-laboratory-protocols

We look forward to receiving your revised manuscript.

Kind regards,

Hector Escriva, PhD

Academic Editor

PLOS ONE

**Journal Requirements:**

2. In your Methods section, please provide additional location information of the collection sites, including geographic coordinates for the data set if available.

4. Thank you for stating that “The funders had no role in study design, data collection and analysis, decision to publish, or preparation of the manuscript” in your financial disclosure.

Please also provide the name of the funders of this study (as well as grant numbers if available) in your financial disclosure statement.

**Comments to the Author**

1. Is the manuscript technically sound, and do the data support the conclusions?

Reviewer #1: Yes

Reviewer #2: Partly

2. Has the statistical analysis been performed appropriately and rigorously? 

Reviewer #1: Yes

Reviewer #2: No

3. Have the authors made all data underlying the findings in their manuscript fully available?

Reviewer #1: No

Reviewer #2: Yes

4. Is the manuscript presented in an intelligible fashion and written in standard English?

Reviewer #1: Yes

Reviewer #2: Yes

5. Review Comments to the Author

Reviewer #1: 1. While I wouldn't ask for repeat of these experiments, given the challenges of working with coral larvae, I'm a little concerned with the conclusions regarding vertical transmission of bacteria from the parental colony. While a previous publication of a co-author does show the inhibitory effects of settlement from the addition of antibiotics, given the annual variation in larvae, there's insufficient evidence to preclude the effects of settlement and metamorphosis resulting from vertically transmitted microbes.

2. While I believe the Mann-whitney test is appropriate for these data, please provide cutoffs in text and in the figures. There's often mention of significance, but statistical values are missing. And although there does appear to be some trends, values are necessary.

3. Please provide values for the statistics on figures, in text, and figure descriptions.

4. Grammatical suggestions, as well as specific points addressed above are provided by line.

Line 45: remove "a" before substantial

Line 46: There are several sentences within the first paragraph (including this line) that are rather long (run-on). Perhaps the paper will read better by separating the sentence? Another instance is Line 58/59.

Line 177: Perhaps it's beneficial to the reader to write out what BCCM/LMG stands for?

Line 184: Please clarify what is being exchange here. I understood it to be bacterial containing seawater, but are filtered fractions from the same batch used for all exchanges over the study period and stored? Or is it a new batch of water from the same source, which is filtered prior to the exchange?

Line 192: insert "of" between "comprised" and "coccoid"

Line 202: missing "the" before "data"

Line 208: remaining  remainder

Line 225: Please indicate the statistical cut-off of significance. Figure S1 also lacks information regarding the cut-off. Also, what exactly is this line referring to? It seems all larvae died by day 12, regardless of the light condition, although those in light:dark cycle at a greater rate.

Line 230: Please provide the statistics. Were there any statistically significant differences in survival rates for data presented in Figure 2?

Line 238: Why is the advanced stage for Figure 3 not categorized as short and long tentacles? Please provide this data. Additionally, please provide statistics for figure 3.

Line 241: rate  rates

Line 241: could be  were

Line 241: It'd be great to have some values provided for this statement regarding survival, as in, what percentage did survive under stagnant conditions?

Line 264: This paragraph refers to light:dark but it's unclear until I read line 314 that light:dark in this section is referring to un-autoclaved SW. Please make this more clear.

Line 273: Is this line referring to figure 5?

Line 285/286: I don't see this is figure 4. Perhaps I should read it as "by" day 35? On day 35 for 2.7 uM-dark, metamorphosis is below 70%.

Line 293: Please provide statistics.

Line 330: Are there any statistics for this?

Given the complexity of the microbiome of the seawater, perhaps some of the observed effects of the fractions were due to competitive interactions between microbes within the seawater resulting in exclusion of pathogens. Therefore, while no "beneficial" bacteria directly interacted with the larvae, pathogenic/harmful bacteria were also prevented from affecting the larvae.

Line 348: insert "of" before 80-90%

Line 349: Is this line referring to Figure S4?

Line 352: remove "the" before "larval ..."

Line 353: I believe this is referring to Figure S5?

Line 356: insert space after "...P14)"

Line 359: Please provide method in more detail, preferably in methods section.

- What does this mean?

Line 361: Planule  planulae

Line 349 - 351: Is this reduction in metamorphosis significant? Also, it seems that although more larvae do achieve the early metamorphosis stage in unchanged water, those under exchanged water conditions achieved advance metamorphosis. Can it really be considered retardation if some larvae are capable of reversion? Also, is the statement in line 352 contradictory to that in line 349-351?

Line 365: Insert "uM" after 0.22

Line 396: I think either "unraveling" or "revealing" was meant to be used here, rather than "unrevealing"

Line 407: As mentioned in the beginning, there's no evidence on what bacteria are vertically transmitted or horizontally acquired. I don't think it's appropriate to discuss acquisition, but rather potential interactions

Line 449: It would have been nice to have a negative control in which larvae were maintained with antibiotics.

Line 450: It's oogenesis and not embryogenesis?

Line 501: There seems to be some information missing after "addition of ..."

The authors investigated seawater filtered through various pore sizes on larval survival and settlement/metamorphosis in order to determine if microbes can have a beneficial role. The authors found survival rates and settlement/metamorphosis of larvae to be partially increased in some instances by the presence of microbes. It's somewhat unclear whether the claims made in the discussion regarding the benefits of particular bacterial species are real. However, there are interesting findings that could certainly provide avenues for further exploration.

Additional questions I had were whether the metamorphic stage of the polyp is a good indication of timing of initiation. Does development to the long tentacle stage occur at the same rate, or can some polyps remain at a short tentacle stage for longer periods?

Reviewer #2: In this manuscript, Freire et al. investigate the effect of bacteria on larval settlement and metamorphosis of the soft coral species Rhytisma fulvum fulvum sampled from the Red Sea. Using an in vitro cultivation system, they assess survival, settlement and metamorphosis performances of wild caught planulae, comparing different types of filtered, autoclaved and bacteria-enriched seawater. This study provides interesting data on octocoral larval settlement, for which little is known. I would suggest few changes in the representation and interpretation of data before publishing the manuscript.

1) In the Material and Method section, the Authors write: “Since data did not fulfill the conditions of normality and homoscedasticity, and could not be improved by transformation, they were analyzed using the non-parametric Wilcoxon-Mann-Whitney test” (line 202). In all graphs, however, spread of data is represented using the mean and standard deviation, generally a bad representation for non-normally distributed data, since they can be strongly affected by extreme values. A more appropriate alternative would be to use median and interquartile range as a measure of dispersion. If mean and standard deviation are to be kept, it should be demonstrated why they provide a good representation of the analyzed data.

2) One of the main conclusions of this study is that vertically inherited bacteria (acquired during the brooding period) are sufficient to induce settlement. This is based on the observation that wild-caught planulae put into sterile (autoclaved) water can still settle and metamorphose, while previous reports showed that planulae put in seawater containing antibiotics do not (Ben-David-Zaslow and Benayahu 1998). From this, the Authors conclude that “vertical transfer of the bacteria to the larvae during oogenesis is sufficient to support larval settlement and development in R. f. fulvum” (line 450). While the Authors are likely correct in their conclusions, experimental data proving this claim are still missing, and it is still possible that environmental bacteria acquired during the course of the experiments were involved in larval settlement. A key control would be to place the planulae first in autoclaved-seawater containing antibiotics and then to replace the medium with autoclaved-seawater, expecting that those planulae would never settle and metamorphose, contrary to those that have not been treated with antibiotics. Considering the difficulty in obtaining new biological material of this species I would not suggest the Authors to perform new experiments. Instead, I would suggest the Authors to be more cautious in the Abstract and Discussion.

3) The Authors make the interesting observation that planulae maintained in constant darkness have initially a higher chance to settle but then lower chance to complete metamorphosis, compared to planulae under a day-night cycle. They conclude that “Light is beneficial for planulae development after day 20 […] indicating the possible presence of beneficial phototrophic bacteria in the coral microbiome” (line 36). However the Authors do not provide any proof for the presence of those phototrophic bacteria and their role in this process. I think the Authors should be more cautious and discuss alternative scenarios – in particular the possibility that the metamorphosing planulae themselves positively respond to light. Many types of photosensitive cells have indeed been described in cnidarians, including at the planula stage in some species.

4) A few typos can be found throughout the text. Few examples (please correct):

Line 27 “in the absence nature reef substrata”

Line 129 “the resultis indicate”

Line 135 “(Forskål 1975)”

I would also suggest changing the title to: “The effect of bacteria on larval settlement and metamorphosis in the octocoral Rhytisma fulvum fulvum”.

6. PLOS authors have the option to publish the peer review history of their article (what does this mean?). If published, this will include your full peer review and any attached files.

Reviewer #1: No

Reviewer #2: No

---

## [Author Response · Author response to Decision Letter 0]

2 Sep 2019

We are enclosing below specific answers to the referee’s queries.

This is a summary of main changes:

1. The title has been changed according to the reviewers’ suggestions.

2. We have included information regarding collections sites and permits obtained to collect the coral planulae. 

3. Following the referees suggestions we have included two new figures in the Supplementary materials, showing the results of the experiments of larvae settlement on plastic and glass plates (2S Fig) and the survival results of experiment performed in 2015 without water exchange (3S Fig). These results were previously quoted as “data not shown” in the manuscript.

4. We have re-dimensioned our claims regarding the effect of specific bacteria in the abstract and discussion.

Referee 1.

We are extremely grateful for the detailed review of the text. We have corrected all the typing and grammar mistakes listed by the referee and provide detailed answers to the referee’s queries below. We fully agree with the referee in that the data presented in the manuscript require further experimentation, including the addition of antibiotics and metagenomic analysis of the larvae in order to allow the achievement of clear conclusions regarding vertical transmission of bacteria. We think that the establishment of this simple culture system will allow carrying out much deeper investigations in the future in this field. We have re-dimensioned our claims regarding vertical transmission of bacteria in the abstract and text. All minor amendments have been done as suggested.

Query: While I believe the Mann-Whitney test is appropriate for these data, please provide cut-offs in text and in the figures. There's often mention of significance, but statistical values are missing. And although there does appear to be some trends, values are necessary. Please provide values for the statistics on figures, in text, and figure descriptions

Answer: We have included the results of the statistical analysis in the text as requested. Due to the high deviations in some of the experiments and the limited number of replicates, many of the observed differences were not statistically significant. The need for a higher number of replicates for this type of experiments was already stated in the last sentence of the first manuscript and we have included an additional comment in this new version (lines 488-491). The limitations in the availability of planula larvae are a problem for cultivation experiments. This new cultivation methodology will allow the establishment of a high number of replicates with a low requirement of coral larvae.

Query: Why is the advanced stage for Figure 3 not categorized as short and long tentacles? Please provide this data. Additionally, please provide statistics for figure 3.

Answer: In the first experiment carried out in 2014 the larvae were divided only in 3 categories. The advanced metamorphosis stage comprises both, short and long tentacles. This fact is explained in the legend of the figure. Unfortunately, no replicates are available for morphology measurements, since in this experiment planulae were pooled together for morphological determination. Therefore, no statistics can be applied to the data represented in Figure 3.

Query: Line 241: It'd be great to have some values provided for this statement regarding survival, as in, what percentage did survive under stagnant conditions?

Answer: We have included a new figure in the Supplementary materials (S3 Fig) that shows the survival under stagnant conditions obtained in the same experiment shown in Fig. 2. We have also included a new figure that shows the results obtained with plastic and glass plates (S2 Fig).

Query: Line 264: This paragraph refers to light:dark but it's unclear until I read line 314 that light:dark in this section is referring to un-autoclaved SW. Please make this more clear.

Answer. We have re-phrased the paragraph. We hope it is clearer now.

Query: Line 285/286: I don't see this is figure 4. Perhaps I should read it as "by" day 35? On day 35 for 2.7 uM-dark, metamorphosis is below 70%.

Answer: Indeed, there was a mistake in the numbers reported in the text. We have amended it. 

Query: Given the complexity of the microbiome of the seawater, perhaps some of the observed effects of the fractions were due to competitive interactions between microbes within the seawater resulting in exclusion of pathogens. Therefore, while no "beneficial" bacteria directly interacted with the larvae, pathogenic/harmful bacteria were also prevented from affecting the larvae.

Answer: Indeed, the observed effects may derive from different interactions in the microbial population: competitive exclusion, probiotic effect or improvement of water quality. As suggested, we have made a more detailed mention of these possibilities in lines 493-496. One of the interesting additions of the work is that the establishment of this experimental system will allow the analysis of changes in the microbial component of the holobiont in future experiments using a very limited number of larvae.

Query: Line 359: Please provide method in more detail, preferably in methods section.

Answer: We have included a more detailed description of this experiment in the methods section.

Query:Line 349 - 351: Is this reduction in metamorphosis significant? Also, it seems that although more larvae do achieve the early metamorphosis stage in unchanged water, those under exchanged water conditions achieved advance metamorphosis. Can it really be considered retardation if some larvae are capable of reversion? Also, is the statement in line 352 contradictory to that in line 349-351?

Answer: We have rephrased this paragraph. The effect of strains P9 and P1 on survival in stagnant cultures is clear and statistically significant (see S4 Fig and S5 Fig). Regarding the effect developmental stage, indeed, reversion is feasible and we have mentioned this in the discussion of the results obtained for the dark maintained cultures (Figure 4), in which the number of total metamorphosed larvae decreased from day 20. This regression is possible for the early metamorphosed stage. We have not observed a decrease in the number of advanced metamorphosed larvae in any of the experiments. Regarding the effect of A. macleodii and Ruegeria mobilis, please note that we do not have data for metamorphosis stages for the 2015 experiment (S5 Fig, former S3 Fig) but in the 2016 experiment, none of the larvae maintained without water exchange entered the advanced metamorphosis stage, but the percentage of attached larvae with A. macleodii P9 (93%) was clearly higher than in the control (40%). 

Query: Line 407: As mentioned in the beginning, there's no evidence on what bacteria are vertically transmitted or horizontally acquired. I don't think it's appropriate to discuss acquisition, but rather potential interactions

Answer: We have rephrased this sentence. The need of additional experiments in order to elucidate the exact mechanisms and composition of the acquired microbiome is stated in lines 500-503, 535-537, 573-576 and in the abstract. 

Query: Line 449: It would have been nice to have a negative control in which larvae were maintained with antibiotics.

Answer: We fully agree with the referee. Nevertheless, the effect of antibiotics on larvae of R. fulvum fulvum has been observed in several independent experiments in the past (Ben-David-Zaslow and Benayahu, 1998), and therefore we are quite confident on this fact.

Query: Additional questions I had were whether the metamorphic stage of the polyp is a good indication of timing of initiation. Does development to the long tentacle stage occur at the same rate, or can some polyps remain at a short tentacle stage for longer periods?

Answer: The evaluation of the developmental stage is tricky, mainly for the initial attachment stages, that can be reversible (see explanation above). It is not possible to follow the individual evolution of the larvae, but as shown for 2016 experiments (Figures 5 and 6), the progression of the number of long-tentacle larvae is continuous with time. 

Reviewer #2:

 In this manuscript, Freire et al. investigate the effect of bacteria on larval settlement and metamorphosis of the soft coral species Rhytisma fulvum fulvum sampled from the Red Sea. Using an in vitro cultivation system, they assess survival, settlement and metamorphosis performances of wild caught planulae, comparing different types of filtered, autoclaved and bacteria-enriched seawater. This study provides interesting data on octocoral larval settlement, for which little is known. I would suggest few changes in the representation and interpretation of data before publishing the manuscript.

Answer. We are grateful for the Reviewer’s comments. We are aware of the fact that additional experiments are required to confirm some of the findings. We think that the fact that we could obtain advanced metamorphosed polyps under such a controlled conditions constitute an interesting tool for future studies. Despite a higher number of replicates will be required in order to obtain statistically significant data, the small number of larvae required for the experiments in this new cultivation methodology is an additional advantage. 

Query: 1) In the Material and Method section, the Authors write: “Since data did not fulfill the conditions of normality and homoscedasticity, and could not be improved by transformation, they were analyzed using the non-parametric Wilcoxon-Mann-Whitney test” (line 202). In all graphs, however, spread of data is represented using the mean and standard deviation, generally a bad representation for non-normally distributed data, since they can be strongly affected by extreme values. A more appropriate alternative would be to use median and interquartile range as a measure of dispersion. If mean and standard deviation are to be kept, it should be demonstrated why they provide a good representation of the analyzed data.

Answer. Indeed, for non-normally distributed data the suggested representation is more suitable, when the number of replicates is high. For measurements with only 3 replicates we think it is not useful. Indeed, more replicates are needed in this type of experiments in order to allow a more reliable statistical analysis. This new experimental system will surely provide a new tool to perform experiments on larval development with a higher number of replicates, despite the general limitation of the number of larvae that can be harvested for research purposes. For now, our main objective was to demonstrate that survival and acceptable development could be achieved in the system, and establishing the basic culture parameters for future experiments. The need for a higher number of replicates for this type of experiments was already stated in the last sentence of the first manuscript and we have included an additional comment in this new version (lines 488-491).

Query: 2) One of the main conclusions of this study is that vertically inherited bacteria (acquired during the brooding period) are sufficient to induce settlement. This is based on the observation that wild-caught planulae put into sterile (autoclaved) water can still settle and metamorphose, while previous reports showed that planulae put in seawater containing antibiotics do not (Ben-David-Zaslow and Benayahu 1998). From this, the Authors conclude that “vertical transfer of the bacteria to the larvae during oogenesis is sufficient to support larval settlement and development in R. f. fulvum” (line 450). While the Authors are likely correct in their conclusions, experimental data proving this claim are still missing, and it is still possible that environmental bacteria acquired during the course of the experiments were involved in larval settlement. A key control would be to place the planulae first in autoclaved-seawater containing antibiotics and then to replace the medium with autoclaved-seawater, expecting that those planulae would never settle and metamorphose, contrary to those that have not been treated with antibiotics. Considering the difficulty in obtaining new biological material of this species I would not suggest the Authors to perform new experiments. Instead, I would suggest the Authors to be more cautious in the Abstract and Discussion.

Answer: We fully agree with the referee in that additional experiments are required in order to elucidate the exact mechanisms of the establishment of the microbial component of the halobiont. As explained above, this fact is stated in lines 500-503, 535-537, 573-576 and in the abstract. 

Query: 3) The Authors make the interesting observation that planulae maintained in constant darkness have initially a higher chance to settle but then lower chance to complete metamorphosis, compared to planulae under a day-night cycle. They conclude that “Light is beneficial for planulae development after day 20 […] indicating the possible presence of beneficial phototrophic bacteria in the coral microbiome” (line 36). However the Authors do not provide any proof for the presence of those phototrophic bacteria and their role in this process. I think the Authors should be more cautious and discuss alternative scenarios – in particular the possibility that the metamorphosing planulae themselves positively respond to light. Many types of photosensitive cells have indeed been described in cnidarians, including at the planula stage in some species.

Answer: Thank you for your comments. Although the differences in development observed between dark and L:D cultures are too high to indicate that light is acting just a “cue” for settlement (Figure 5), we fully agree in that additional experiments are required in order to confirm that the difference is due to the metabolic interactions with phototrophic symbiont bacteria. We have rephrased this section. A more detailed list of the different possible explanations for the increase in larval metamorphosis in the presence of light despite being devoid of the photosynthetic algal symbiont has been provided (lines 515-517).

Query: I would also suggest changing the title to: “The effect of bacteria on larval settlement and metamorphosis in the octocoral Rhytisma fulvum fulvum”.

Answer: We have changed the title as suggested.

---

## [Decision Letter · Decision Letter 1]

17 Sep 2019

The effect of bacteria on planula-larvae settlement and metamorphosis in the octocoral Rhytisma fulvum fulvum

PONE-D-19-17240R1

Dear Dr. Otero,

We are pleased to inform you that your manuscript has been judged scientifically suitable for publication and will be formally accepted for publication once it complies with all outstanding technical requirements.

With kind regards,

Hector Escriva, PhD

Academic Editor

PLOS ONE

Additional Editor Comments (optional):

Reviewers' comments:

Reviewer's Responses to Questions

**Comments to the Author**

1. If the authors have adequately addressed your comments raised in a previous round of review and you feel that this manuscript is now acceptable for publication, you may indicate that here to bypass the “Comments to the Author” section, enter your conflict of interest statement in the “Confidential to Editor” section, and submit your "Accept" recommendation.

Reviewer #1: All comments have been addressed

Reviewer #2: All comments have been addressed

2. Is the manuscript technically sound, and do the data support the conclusions?

Reviewer #1: Yes

Reviewer #2: Yes

3. Has the statistical analysis been performed appropriately and rigorously? 

Reviewer #1: Yes

Reviewer #2: Yes

4. Have the authors made all data underlying the findings in their manuscript fully available?

Reviewer #1: Yes

Reviewer #2: No

5. Is the manuscript presented in an intelligible fashion and written in standard English?

Reviewer #1: Yes

Reviewer #2: Yes

6. Review Comments to the Author

Reviewer #1: The authors addressed the issues brought up by the reviewers. While they acknowledge the issues in replicate numbers, the previous interpretations of these data have been addressed. As the authors point out, these findings can inform future experiments and the manuscript should be accepted for publication.

Reviewer #2: The Authors have appropriately addressed all the reviewer comments.

Related to 4. - According to the PLOS Data policy, the data point values should be made available. I could not find this information in the files provided by the Authors despite their claim that "All relevant data are within the manuscript and its Supporting Information files."

7. PLOS authors have the option to publish the peer review history of their article (what does this mean?). If published, this will include your full peer review and any attached files.

Reviewer #1: No

Reviewer #2: No

---

## [Editor Report · Acceptance letter]

19 Sep 2019

PONE-D-19-17240R1 

The effect of bacteria on planula-larvae settlement and metamorphosis in the octocoral *Rhytisma fulvum fulvum*

Dear Dr. Otero:

I am pleased to inform you that your manuscript has been deemed suitable for publication in PLOS ONE. Congratulations! Your manuscript is now with our production department. 

With kind regards,

on behalf of

Dr. Hector Escriva 

Academic Editor

PLOS ONE